# Cryo-EM structure of an amyloid fibril formed by full-length human SOD1 reveals its conformational conversion

Li-Qiang Wang [1,2,11], Yeyang Ma [3,4,11], Han-Ye Yuan [1,2,11], Kun Zhao [3,4], Mu-Ya Zhang [1,2], Qiang Wang [5], Xi Huang [6], Wen-Chang Xu[1], Bin Dai[7], Jie Chen[1,2], Dan Li [8,9], Delin Zhang [5], Zhengzhi Wang [10], Liangyu Zou [6], Ping Yin [5], Cong Liu [3✉] & Yi Liang [1,2✉]

Amyotrophic lateral sclerosis (ALS) is a neurodegenerative disease. Misfolded Cu, Zn-superoxide dismutase (SOD1) has been linked to both familial and sporadic ALS. SOD1 fibrils formed in vitro share toxic properties with ALS inclusions. Here we produced cytotoxic amyloid fibrils from full-length apo human SOD1 under reducing conditions and determined the atomic structure using cryo-EM. The SOD1 fibril consists of a single protofilament with a left-handed helix. The fibril core exhibits a serpentine fold comprising N-terminal segment (residues 3–55) and C-terminal segment (residues 86–153) with an intrinsic disordered segment. The two segments are zipped up by three salt bridge pairs. By comparison with the structure of apo SOD1 dimer, we propose that eight β-strands (to form a β-barrel) and one α-helix in the subunit of apo SOD1 convert into thirteen β-strands stabilized by five hydrophobic cavities in the SOD1 fibril. Our data provide insights into how SOD1 converts between structurally and functionally distinct states.

[1] Hubei Key Laboratory of Cell Homeostasis, College of Life Sciences, Wuhan University, 430072 Wuhan, China. [2] Wuhan University Shenzhen Research Institute, 518057 Shenzhen, China. [3] Interdisciplinary Research Center on Biology and Chemistry, Shanghai Institute of Organic Chemistry, Chinese Academy of Sciences, 201210 Shanghai, China. [4] University of Chinese Academy of Sciences, 100049 Beijing, China. [5] National Key Laboratory of Crop Genetic Improvement and National Centre of Plant Gene Research, Huazhong Agricultural University, 430070 Wuhan, China. [6] Department of Neurology, the Second Clinical Medical College, Jinan University (Shenzhen People's Hospital), 518020 Shenzhen, China. [7] Institute of Nano Biomedicine and Engineering, Department of Instrument Science and Engineering, School of Electronic Information and Electrical Engineering, Shanghai Jiao Tong University, Shanghai, China. [8] Key Laboratory for the Genetics of Developmental and Neuropsychiatric Disorders, Ministry of Education, Bio-X Institutes, Shanghai Jiao Tong University, 200030 Shanghai, China. [9] Zhangjiang Institute for Advanced Study, Shanghai Jiao Tong University, 200240 Shanghai, China. [10] School of Civil Engineering, Wuhan University, 430072 Wuhan, China. [11] These authors contributed equally: Li-Qiang Wang, Yeyang Ma, Han-Ye Yuan. ✉email: liulab@sioc.ac.cn; liangyi@whu.edu.cn

A myotrophic lateral sclerosis (ALS), also called Lou Gehrig's disease, is a progressive, fatal neurodegenerative disease that involves the loss of upper and lower motor neurons[1–4]. Ninety percent of ALS cases are sporadic and little is known about the origin, while ten percent of ALS cases are inherited familial ALS[1–7]. The *sod1* gene, serving as a major antioxidant gene, was the first to be linked to the familial form of ALS[8] and other genes associated with genetic ALS include those encoding TDP-43 and FUS[1,6]. The misfolding of human Cu, Zn-superoxide dismutase (SOD1) in motor neuron cells plays a crucial role in the etiology of the disease[1,3,4,6,7,9]. Misfolded SOD1 aggregates were widely observed in the spinal cords of both genetic ALS and sporadic ALS cases[3,4,7]. The functional human SOD1 is a 32-kDa homo-dimeric metalloenzyme; each subunit consists of 153 amino acids and contains one copper ion and one zinc ion[10,11]. The SOD1 structure in each subunit features an antiparallel β-barrel composed of eight β-strands and two α-helices, which is stabilized by a disulfide bond between Cys57 and Cys146 (refs. [10,11]). In sharp contrast, the high-resolution structures of SOD1 amyloid fibrils are not available so far[1,12–14]. Therefore, it is unclear for the conformational conversion of SOD1 from its immature form with no post-translational modifications into an aggregated form during the pathogenesis of ALS.

The mature form of SOD1 is exceptionally stable and it is very unlikely that the mature, metalated, dimeric, and disulfide-intact form ever converts into the aggregated form[15–17]. Instead, it has been proposed that immature forms of SOD1, which lack copper and zinc ions and the disulfide bond in the structure, are the origin for cytotoxic misfolded conformations[15–19]. Previous studies have shown that the amyloid-like aggregates isolated from ALS transgenic mice or cells expressing ALS-causing SOD1 mutations contain metal-deficient and disulfide-reduced SOD1, suggesting their pathogenic potential[20–22]. Substantial experiments have demonstrated that the full-length apo SOD1 can convert into amyloid fibrils by incubation with reducing agents in vitro[23–30]. Importantly, SOD1 fibrils formed in vitro are able to incorporate into cells and transmit intercellularly[7]. Moreover, SOD1 fibrils produced under reducing conditions also share pathological properties with ALS inclusions, such as the ability to induce mitochondria damage, cause neuroinflammation and activate microglial cells triggering neurodegeneration in ALS[31–33]. Thus, structural determination of the SOD1 fibril is of importance for understanding the pathogenic mechanism of SOD1 in both familial and sporadic ALS.

Here we prepared homogeneous cytotoxic amyloid fibrils from recombinant, full-length apo human SOD1 under reducing conditions and determine the atomic structure by using cryo-EM. Our findings provide structural insights into the conversion of SOD1 between physiological and fibrillar states.

## Results

**SOD1 forms cytotoxic amyloid fibrils under reducing conditions**. Treatment of the apo form of SOD1 with 5–10 mM tris (2-carboxyethyl) phosphine (TCEP), a highly stable disulfide-reducing agent, generates a reduced state that is able to mimic physiological reducing environments. We produced amyloid fibrils from recombinant, full-length apo human SOD1 (residues 1 to 153) overexpressed in *Escherichia coli*, by incubating the purified apoprotein in 20 mM tris-HCl buffer (pH 7.4) containing 5 mM TCEP and shaking at 37 °C for 40–48 h (see methods). SOD1 fibrils formed under such reducing conditions were concentrated to ~30 μM in a centrifugal filter (Millipore) and examined by electron microscopy without further treatment.

Negative-staining transmission electron microscopy (TEM) imaging showed that recombinant, full-length apo SOD1 formed homogeneous and unbranched fibrils under reducing conditions (Supplementary Fig. 1a). The SOD1 fibril is composed of a single protofilament with a fibril full width of $11.3 \pm 1.0$ nm ($n = 8$) (Supplementary Fig. 1a), which is consistent with previously described in vitro amyloid fibrils produced from full-length apo SOD1 under the same reducing conditions, which showed a width of ~11 nm based on negative staining on TEM[29]. Congo red binding assays showed a red shift of the maximum absorbance, from 490 to 550 nm, in the presence of SOD1 fibrils (Supplementary Fig. 1b), which is the characteristics of amyloid fibrils[34,35]. Notably, the SOD1 fibrils exhibited cytotoxicity to both SH-SY5Y cells (Supplementary Fig. 1c, d) and HEK-293T cells (Supplementary Fig. 1e, f) in a dose-dependent manner, and caused severe mitochondrial impairment in both cell lines (Supplementary Fig. 2a–j). Together, these data showed that full-length apo SOD1 forms cytotoxic, mitochondrial dysfunction-inducing amyloid fibrils under reducing conditions.

**Cryo-EM structure of SOD1 fibril**. We next determined the atomic structure of the cytotoxic SOD1 amyloid fibrils by cryo-EM (Table 1 and Figs. 1 and 2). The cryo-EM micrographs, two-dimensional (2D) class average images, and atomic force microscopy (AFM) images show that the SOD1 fibril is composed of a single protofilament with a left-handed helical twist (Fig. 1a, Supplementary Fig. 3a, and Supplementary Fig. 4a–e). The helical pitch is $144 \pm 5$ nm (Fig. 1a) or $146 \pm 5$ nm (Supplementary Fig. 4a–e). The SOD1 subunit within the protofilament is arranged in a staggered manner (Supplementary Fig. 3b). The fibrils are morphologically homogeneous, showing a fibril full width of $12.3 \pm 0.7$ nm (Fig. 1a and Supplementary Fig. 3a) or $12.9 \pm 1.0$ nm (Supplementary Fig. 4a–e).

Using helical reconstruction in RELION3.1 (ref. [36]), we determined a density map of the ordered core of SOD1 fibril, with an overall resolution of 2.95 Å, which features well-resolved side-chain densities and clearly separated β strands along the fibril axis (Fig. 1b and Supplementary Fig. 5). Cross-sectional view of the 3D map of the SOD1 fibril and top view of the density map show a protofilament comprising the N- and the C-terminal segments, with an unstructured flexible region in between (Fig. 1b, e). The 3D map showed a single protofilament in the SOD1 fibril with a left-handed helix, and the left-handed structure of the fibril is supported by AFM images (Supplementary Fig. 4a–e). The half-helical pitch is 73.1 nm (Fig. 1c). The SOD1 subunit within the protofilament stacks along the fibril axis with a helical rise of 4.82 Å and a twist of $-1.187°$ (Fig. 1d).

We unambiguously built a structure model of SOD1 fibril comprising the N-terminal segment (residues 3 to 55) and the C-terminal segment (residues 86 to 153) at 2.95 Å (Fig. 2). The density of an intrinsic disordered segment comprising residues 56 to 85 is invisible due to high flexibility (Fig. 2a–c), which is reminiscent of the internal disordered segments observed in the structures of patient-derived amyloid fibrils from systemic AL amyloidosis[37,38]. The presence of the internal disordered segment represents an interesting structural feature of SOD1 fibril formed under reducing conditions.

Side chains for the residues in the SOD1 fibril core can be well accommodated into the density map (Fig. 2a). The exterior of the SOD1 fibril core is mostly hydrophilic, whereas side chains of most hydrophobic residues are located in the interior of the SOD1 fibril fold (Fig. 2b–g and Supplementary Fig. 6). Five hydrophobic cavities (Supplementary Fig. 6a and Fig. 2g), four hydrogen bonds (Supplementary Fig. 7a), and a very compact

**Table 1 Cryo-EM data collection, refinement, and validation statistics.**

| | SOD1 fibril (EMD-32227, PDB 7VZF) |
|---|---|
| *Data collection and processing* | |
| Magnification | 130,000 |
| Voltage (kV) | 300 |
| Camera | K2 summit (Titan Krios) |
| Frame exposure time (s) | 0.16 |
| Movie frames ($n$) | 40 |
| Electron exposure (e⁻/Å²) | 60 |
| Defocus range (μm) | −2.0 to −1.2 |
| Pixel size (Å) | 1.04 |
| Symmetry imposed | C1 |
| Box size (pixel) | 320 |
| Inter-box distance (Å) | 33.3 |
| Micrographs collected ($n$) | 2931 |
| Segments extracted ($n$) | 288,744 |
| Segments after Class2D ($n$) | 147,525 |
| Segments after Class3D ($n$) | 70,067 |
| Map resolution (Å) | 2.95 |
| FSC threshold | 0.143 |
| Map resolution range (Å) | 2.30–5.01 |
| Refinement | |
| Initial model used | De novo |
| Model resolution (Å) | 2.95 |
| FSC threshold | 0.143 |
| Model resolution range (Å) | 2.95 |
| Map sharpening $B$ factor (Å²) | −77.93 |
| Model composition | |
| Nonhydrogen atoms | 2,628 |
| Protein residues | 363 |
| Ligands | 0 |
| $B$ factors (Å²) | |
| Protein | 70.90 |
| R.m.s. deviations | |
| Bond lengths (Å) | 0.009 |
| Bond angles (°) | 1.060 |
| Validation | |
| MolProbity score | 2.86 |
| Clashscore | 37.29 |
| Poor rotamers (%) | 0 |
| Ramachandran plot | |
| Favored (%) | 73.50 |
| Allowed (%) | 26.50 |
| Disallowed (%) | 0 |

fold (Fig. 2b, d) help stabilize the fibril core, as described in detail below.

Hydrophobic side chains of Val5, Leu8, Val14, and Ile17, hydrophobic side chains of Ile18, Phe20, Leu42, Phe45, Val47, and Phe50, and hydrophobic side chains of Val29, Leu31, Ile35, and Leu38 are located in the interior of the N-terminal part of SOD1 fibril to form three hydrophobic cavities (Supplementary Fig. 6b–d), and hydrophobic side chains of Val87, Ala89, and Ala95 and hydrophobic side chains of Val94, Val97, Ile99, Ile104, Leu106, Ile113, Leu144, Ala145, Val148, Ile149, Ile151, and Ala152 are located in the interior of the C-terminal part to form two hydrophobic cavities (Supplementary Fig. 6e, f), thereby stabilizing the SOD1 fibril. Two pairs of amino acids (Val14 and Asn53; and Gln15 and Phe50) from the N-terminal segment (Supplementary Fig. 7b) and two pairs of amino acids (Gly130 and Asp125; and Arg143 and Gln153) from the C-terminal segment (Supplementary Fig. 7c, d) form four hydrogen bonds to stabilize the fibril core.

The SOD1 fibril core features a very compact fold containing thirteen β-strands (β1 to β13) and an in-register intramolecular β-strand architecture (Fig. 2b, d). Six β-strands (β1 to β6) and seven β-strands (β7 to β13) are present in the N- and C-terminal segments of the SOD1 fibril core structure, respectively (Fig. 2b–d). The height of one layer along the helical axis is 15.82 Å, which is the distance between the highest point in the loop between β4 and β5 and the lowest point in the loop between β8 and β9 (Fig. 2e).

The SOD1 fibril contains a long intramolecular interface comprising residues 36 to 48 in the N-terminal half and residues 98 to 109 in the C-terminal half (Fig. 3a). Three pairs of intramolecular salt bridges formed by Lys36 and Asp109, His43 and Asp101, and His46 and Glu100 (with distances <4 Å; Fig. 3b–e) are identified to stabilize the intramolecular L-shaped interface between the N- and C-terminal parts of SOD1 fibril (Figs. 2a, b, f, g and 3b–e). Side chains of most residues (Lys36, Thr39, His43, His46, His48, Ser98, Glu100, Asp101, Ser105, Ser107, and Asp109) in the interior of the intramolecular L-shaped interface are hydrophilic (Figs. 2b, g and 3a). The presence of a mostly hydrophilic intramolecular interface represents another interesting structural feature of SOD1 fibril formed under reducing conditions.

## Discussion

SOD1 is involved in the pathogenesis of the motor neuron disease ALS where it is observed to form intracellular fibrillar inclusions[3,7,39,40]. These proteinaceous inclusions are also observed in cases of Parkinson's disease and aged individuals[41,42]. Here, we presented cryo-EM structure of a human SOD1 fibril and compared the structures of apo SOD1 dimer and SOD1 fibril produced under reducing conditions (Fig. 4). Notably, the SOD1 molecule adopts largely distinctive secondary, tertiary, and quaternary structures in two different states of SOD1, highlighting the high structural polymorphs and phenotypic diversity of SOD1 in physiological and fibrillar states. The apo human SOD1 dimer contains eight β-strands (to form an antiparallel β-barrel), two α-helices, and a single disulfide bond between Cys57 in α1 and Cys146 in β8' in each subunit as well as an intermolecular interface involving strong hydrophobic interactions and hydrogen bonding from Gly51 and Gly114 of one molecule to Ile151 of the other[10] (Fig. 4a, b). In contrast, once folding into cytotoxic, mitochondrial dysfunction-inducing fibril structure, SOD1 molecules form six β-strands (β1 to β6) by its N-terminal segment (residues 3 to 55) and seven β-strands (β7 to β13) by its C-terminal segment (residues 86 to 153), exhibiting an in-register intramolecular β strand architecture (Fig. 4a, c). Moreover, the cytotoxic SOD1 fibril structure features a long, mostly hydrophilic intramolecular L-shaped interface and an intrinsic disordered segment comprising residues 56 to 85 (Fig. 4a, c). Once apo SOD1 dimer converts into its fibrillar form, the SOD1 molecule undergoes a completely conformational rearrangement, with the antiparallel β-barrel of apo SOD1 converted to β1–β5, β7, β8, β11, and β13, the loop between β4' and α1 converted to β6, α1 and the loop between α1 and β5' converted to the internal disordered segment, the loop between β6' and β7' converted to β9 and β10, and α2 of apo SOD1 converted to β12 in the SOD1 fibril (Fig. 4b, c). Of note, the pathological relevance of the SOD1 fibril reported here remains unknown, although treatment of cells with the fibrils disrupts mitochondrial membrane permeability and integrity[31], causes severe mitochondrial impairment (this work) and inflammation[32], and activates microglia[33]. It is still unclear whether the fibrils are pathogenic agents in ALS[9,13].

This work builds on previous intensive studies of SOD1 fibril formation[12,14,19,23–30,43]. Valentine and colleagues extensively investigated SOD1 fibrils and used EM and AFM to characterize the fibrils formed by SOD1 and its mutants[23,26]. Shaw and co-

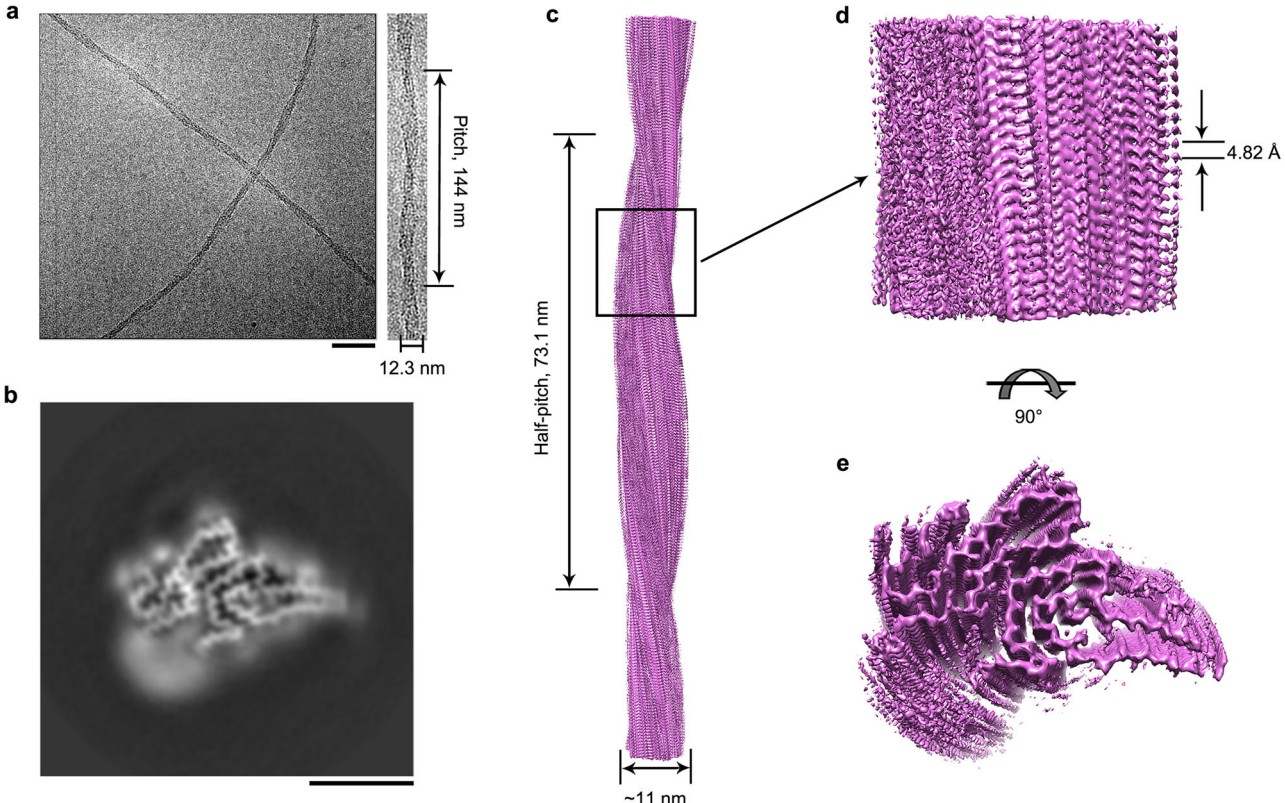

**Fig. 1 Cryo-EM structure of SOD1 fibril. a** Raw cryo-EM image of amyloid fibrils assembled from recombinant, full-length apo human SOD1 under reducing conditions. Enlarged section of **a** (right) showing a single protofilament intertwined into a left-handed helix, with a fibril full width of 12.3 ± 0.7 nm and a helical pitch of 144 ± 5 nm. The scale bar represents 50 nm. The helical pitch and fibril width were measured and expressed as the mean ± SD of values obtained in $n = 8$ biologically independent measurements. Source data are provided as a Source Data file. **b** Cross-sectional view of the 3D map of the SOD1 fibril showing a protofilament comprising the N-terminal part (left) and the C-terminal part (right), with an unstructured flexible region (bottom). Scale bars, 5 nm. **c** 3D map showing a single protofilament (in orchid) intertwined into a left-handed helix, with a fibril core width of ~11 nm and a half-helical pitch of 73.1 nm. **d** Enlarged section showing a side view of the density map. Close-up view of the density map in **c** showing that the subunit in protofilament stacks along the fibril axis with a helical rise of 4.82 Å. **e** Top view of the density map.

workers also studied SOD1 fibrils and showed the consistent formation of the fibrils under reducing conditions[30]. The Hart laboratory determined the crystal structures of two pathogenic SOD1 mutants S134N and H46R, termed metal-binding region mutants, and proposed a mechanism of amyloid structures assembled from S134N (or H46R) dimers[43]. The Eisenberg group reported the crystal structures of several key fibril-forming segments of SOD1[14].

Previous studies proposed two alternative models of SOD1 fibril based on protease digestion experiments and mass spectrometric analyses[25,26]. The so-called "three key region model" predicts that the SOD1 fibril core contains one N-terminal segment comprising residues 1 to 30 and two C-terminal segments comprising residues 90–120 and 135–153 (ref. [25]). This is in good agreement with our model, wherein β1 and β2, β8–β11, and β13 would correspond to the first, second, and third segments in the three key region model[25]. The other N-terminal core model predicts that the SOD1 fibril core contains the first 63 residues of the N terminus of the protein[29]. This is partly compatible with our model, wherein the six β strands (β1 to β6) present in the N-terminal segment would correspond to the minimal protease-resistant core region comprising residues 1–63 in the N-terminal core model[26]. In all three models, SOD1 fibrils are produced from the immature form of the protein under reducing conditions. Previous work had shown that amyloid fibril formation is initiated by the immature, disulfide-reduced, apo form of SOD1

(ref. [23]). In our SOD1 fibril model, Cys57, Cys111, and Cys146 are all in disulfide-reduced conformations with free thiol groups, and misfolded and aggregated SOD1 evolves from a pool of immature SOD1. Intriguingly, the side chains of those Cys residues appear exposed to quite crowded areas of the first and fifth hydrophobic cavities (Fig. 2a, b and Supplementary Fig. 6b, f).

Strikingly, among two hundred and sixteen genetic mutations of SOD1 identified from different familial ALS[1–5,8,17–22,25,26,29,44–50] (https://alsod.iop.kcl.ac.uk/), one hundred and eighty-two clinically identified mutations are located within the SOD1 fibril core structure determined in this study, in which one hundred and five representative genetic ALS-related mutations are listed in Fig. 4a. Notably, residues forming strong salt bridges (His43, His46, Glu100, Asp101, and Asp109) that contribute to the stabilization of the intramolecular L-shaped interface between the N- and C-terminal parts of SOD1 fibril (Figs. 2a, b, f, g and 3b–e) or hydrogen bonds (Val14 and Asp125) that contribute to the maintenance of the SOD1 fibril structure (Supplementary Fig. 7b–d) are also ALS-associated mutation sites[1–5,8,17–22,25,26,29,44–50]. Based on the cryo-EM fibril structure, the disease mutations, such as H43R, H46R, H46D, E100G, E100K, D101G, D101H, D101N, D101Y, D109Y, D109N, and D125H (Fig. 4a, salt bridge mutations, and hydrogen bond mutations), may disrupt important interactions in the cytotoxic SOD1 fibril structure. This suggests that the different mutations may induce SOD1 to form fibrils with structures and cytotoxicity distinct from the one presented here, which might be

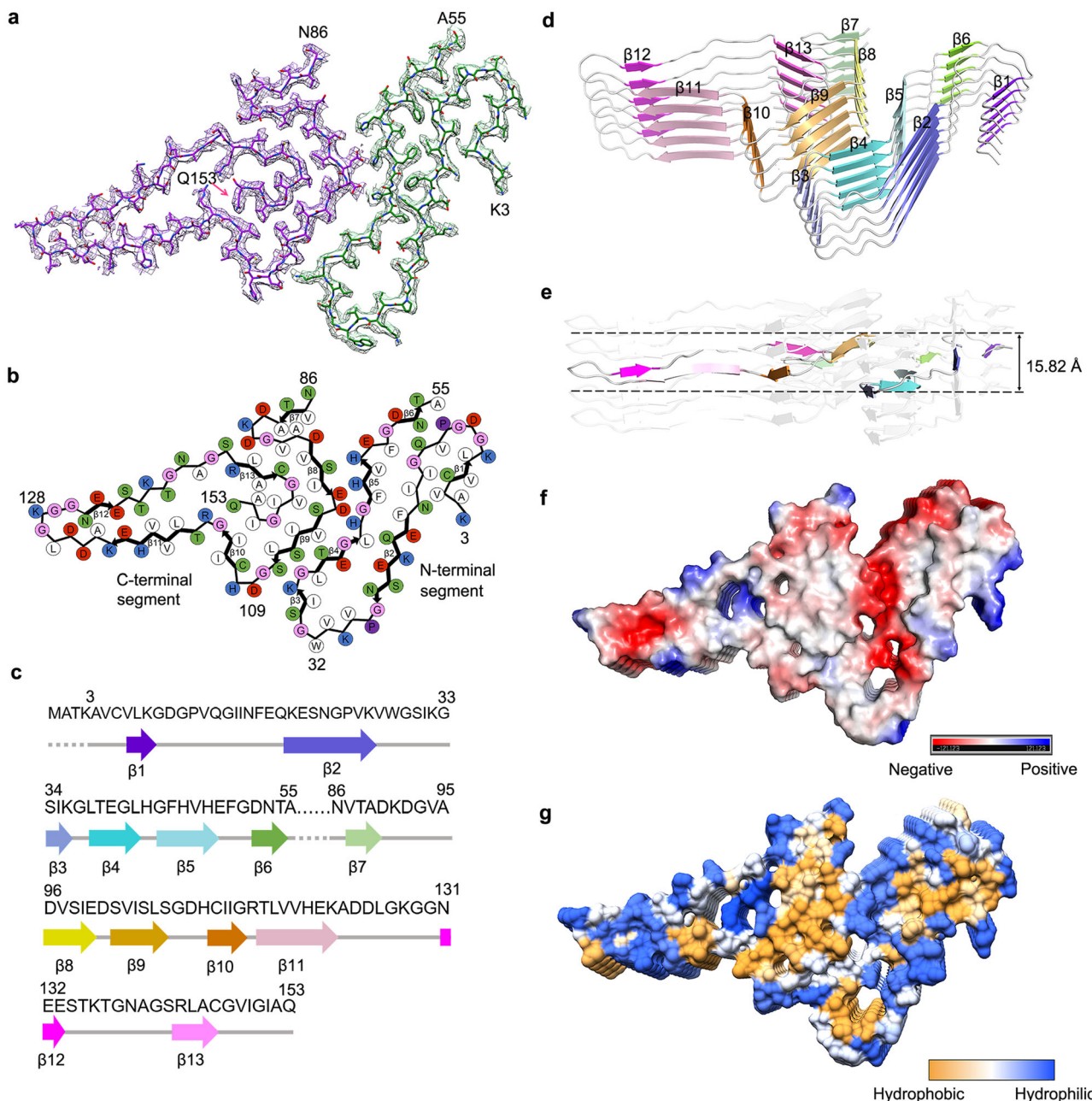

**Fig. 2 Atomic structure of SOD1 fibril. a** Cryo-EM map of an amyloid fibril from recombinant, full-length apo human SOD1 with the atomic model overlaid. The SOD1 fibril core comprises the N-terminal segment (residues 3–55) and the C-terminal segment (residues 86 to 153) colored green and purple, respectively, with an internal disordered segment. **b** Schematic view of the SOD1 fibril core. Residues are colored as follows: white, hydrophobic; green, polar; red and blue, negatively and positively charged, respectively; and magenta, glycine. β strands are indicated with bold lines. Side chains of most hydrophobic residues are located in the interior of the SOD1 fibril fold. **c** Sequence of the fibril core comprising residues 3–55 and 86–153 from full-length human SOD1 (1 to 153) with the observed six β strands colored violet (β1), blue (β2), light blue (β3), cyan (β4), light cyan (β5), and green (β6) in the N-terminal region and the observed seven β strands colored light green (β7), yellow (β8), gold (β9), orange (β10), pink (β11), magenta (β12), and light magenta (β13) in the C-terminal region. The dotted lines correspond to residues 1–2 and residues 56–85 not modeled in the cryo-EM density. **d** Ribbon representation of the structure of an SOD1 fibril core containing five molecular layers and two segments. **e** As in **d**, but viewed perpendicular to the helical axis, revealing that the height of one layer along the helical axis is 15.82 Å. **f** Electrostatic surface representation of the structure of an SOD1 fibril core containing five molecular layers and two segments. **g** Hydrophobic surface representation of the structure of an SOD1 fibril core as in **d**. **f**, **g** Three pairs of amino acids (Lys36 and Asp109; His43 and Asp101; and His46 and Glu100) form three salt bridges at the intramolecular interface between the N- and C-terminal regions of SOD1 fibril. The surface of two regions of the SOD1 fibril core is shown according to the electrostatic properties (red, negatively charged; blue, positively charged) (**f**) or the hydrophobicity (yellow, hydrophobic; blue, hydrophilic) (**g**) of the residues.

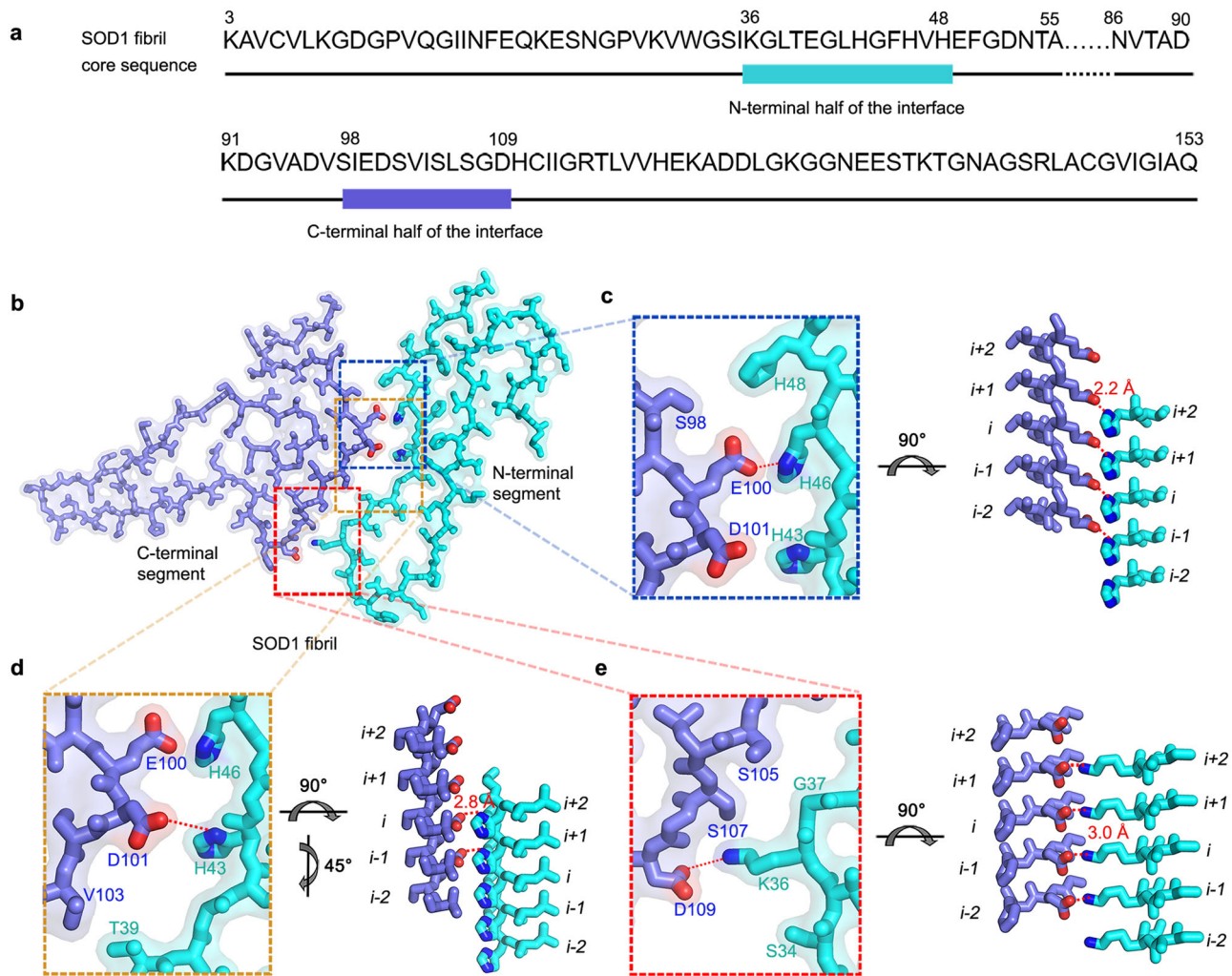

**Fig. 3 Close-up view of the intramolecular interface between the N- and C-terminal regions of SOD1 fibril. a** The primary sequence of the SOD1 fibril core. The cyan bar marks the N-terminal half of the interface and the blue bar marks the C-terminal half. The dotted lines correspond to residues 56 to 85 not modeled in the cryo-EM density and represent an internal disordered segment between the N- and C-terminal regions of SOD1 fibril. **b** A space-filled model overlaid onto stick representation of the SOD1 fibril, in which the N-terminal segment is shown in cyan and the C-terminal segment in blue. Lys/Asp pairs, His/Asp pairs, and His/Glu pairs that form salt bridges are highlighted in red (oxygen atoms in Asp and Glu) and blue (nitrogen atom in Lys and His), and the intramolecular interface is magnified in **c** to **e**. **c–e** Magnified top views of the three regions of the intramolecular L-shaped interface between the N- and C-terminal regions of SOD1 fibril, where three pairs of amino acids (Lys36 and Asp109; His43 and Asp101; and His46 and Glu100) form three salt bridges. Two side views (right) highlighting a strong salt bridge between Glu100 from the C-terminal segment (*i*) and His46 from the adjacent N-terminal segment (*i* + 1), with a distance of 2.2 Å (red), or between Asp101 from the molecular layer (*i*–1) of the C-terminal region and His43 from the molecular layer (*i* + 2) of the N-terminal region, with a distance of 2.8 Å (red). A side view (right) highlighting a strong salt bridge between Asp109 from the C-terminal part (*i*) and Lys36 from the adjacent N-terminal part (*i* + 1), with a distance of 3.0 Å (red).

related to the structural diversity of SOD1 fibrils, strains, and phenotypic diversity of SOD1 in pathological state[1,23,25]. Thus, we plan to collect structural data on various SOD1 mutations including metal-binding region mutants H46R, H46D, G85R, and D125H and wild-type-like mutants A4V, D90A, and G93A in the near future. Interestingly, as for the ALS-associated residues including His43, His46, Glu100, Asp101, and Asp109, His43 forms a hydrogen bond with Thr39, His46 forms a strong salt bridge with Asp124, and Asp101 forms a strong salt bridge with Arg79 in the subunit to stabilize the structure of apo SOD1 dimer (Supplementary Fig. 8a–d), whereas His43 and Asp101 form a strong salt bridge and His46 forms a strong salt bridge with Glu100 in the SOD1 fibril to stabilize the intramolecular L-shaped interface (Fig. 3b–d). This indicates that reorganization of salt bridges may occur for these ALS-associated mutated SOD1 during their conformational conversion from apo into fibrillar form.

In summary, we revealed by cryo-EM that the full-length human SOD1 displays a novel amyloid fibril structure. The SOD1 fibril displays a very compact fold with an internal disordered segment, which contains thirteen β-strands stabilized by five hydrophobic cavities and four hydrogen bonds, and a long, mostly hydrophilic intramolecular L-shaped interface stabilized by three strong salt bridges. The comparison of the structures of apo SOD1 dimer and SOD1 fibril reveals the substantial conformational conversion from a β-sheet-rich (correspond to the antiparallel β-barrel structure), immature form of SOD1 to a totally distinct β-sheet-rich (correspond to an in-register intramolecular β strand architecture), fibrillar form of SOD1 during pathogenesis of ALS. The fibril structure will be valuable in regard to understanding the structural basis underlying SOD1 misfolding and inspiring future research on the structural polymorphism of SOD1 strains and their relationship to ALS.

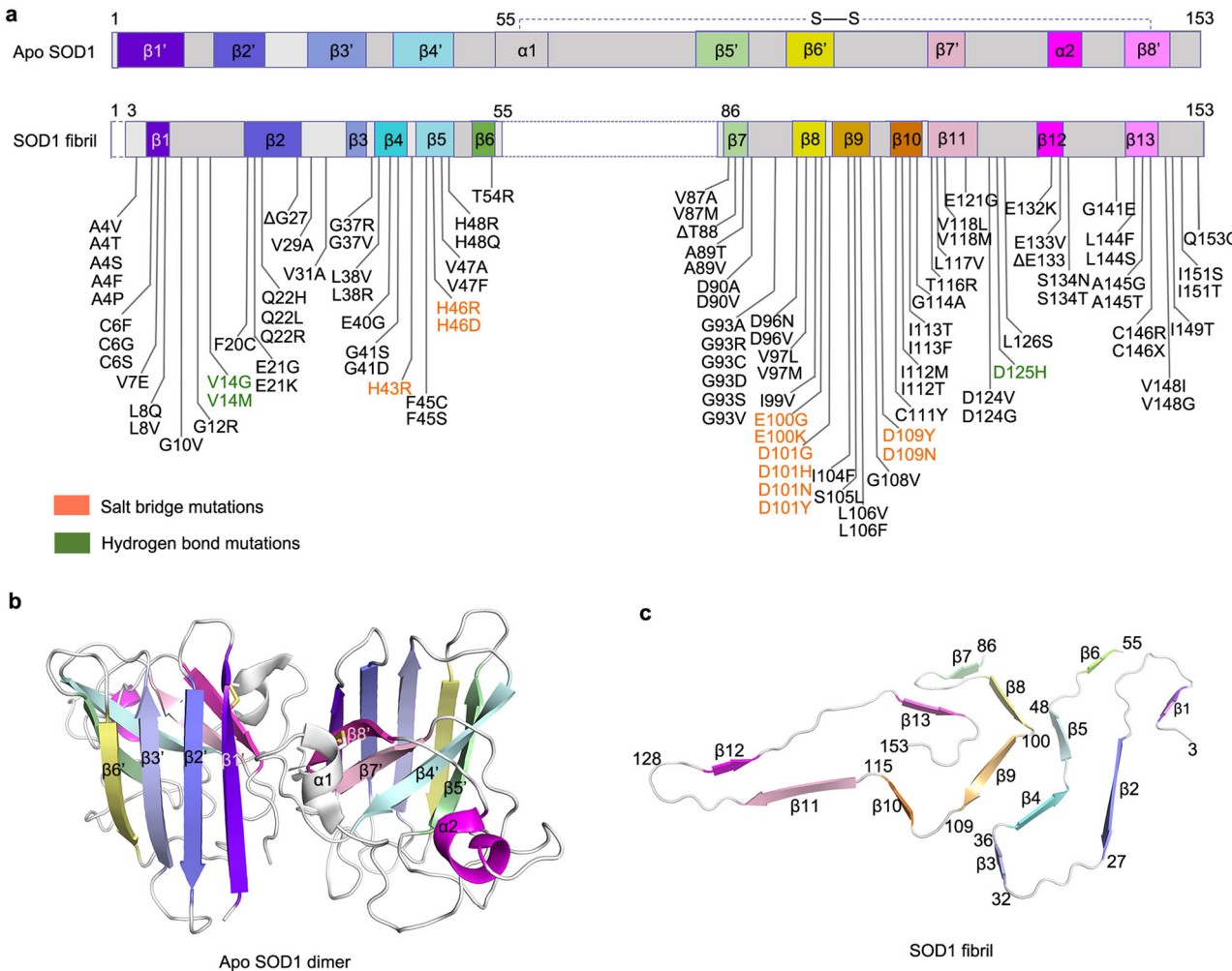

**Fig. 4 Comparison of the structures of apo SOD1 dimer and SOD1 fibril. a** Sequence alignment of the full-length apo human SOD1 (1 to 153) monomer with eight β-strands colored violet (β1'), blue (β2'), light blue (β3'), light cyan (β4'), light green (β5'), yellow (β6'), pink (β7'), and light magenta (β8'), two α-helices colored gray (α1) and magenta (α2), and a single disulfide bond between Cys[57] in α1 and Cys[146] in β8' in each subunit[10]. Sequence alignment of the SOD1 fibril core comprising residues 3–55 and 86–153 from full-length human SOD1 (1 to 153) with the observed thirteen β strands colored violet (β1), blue (β2), light blue (β3), cyan (β4), light cyan (β5), green (β6), light green (β7), yellow (β8), gold (β9), orange (β10), pink (β11), magenta (β12), and light magenta (β13). Dashed boxes correspond to residues 1–2 and residues 56–85, which were not modeled in the cryo-EM density. Among two hundred and sixteen mutations linked to familial ALS[1–5,8,17–22,25,26,29,44–50] (https://alsod.iop.kcl.ac.uk/), one hundred and eighty-two clinically identified mutations are located within the SOD1 fibril core structure, in which one hundred and five representative genetic ALS-related mutations are listed in **a**. Orange, salt bridge mutations; green, hydrogen bond mutations. **b** Ribbon representation of the structure of full-length apo human SOD1 (1–153) dimer with eight β-strands (to form a β-barrel), two α-helices, and a disulfide bridge (yellow line) linking α1 and β8' in each subunit (PDB 1HL4)[10]. **c** Ribbon representation of the structure of an SOD1 fibril core containing one molecular layer and thirteen β-strands, in which the N-terminal segment contains six β strands (β1–β6) and the C-terminal segment contains seven β strands (β7–β13).

## Methods

**Protein purification**. A plasmid-encoding, full-length human SOD1 (1–153) was a gift from Dr. Thomas O'Halloran (Chemistry of Life Processed Institute, Northwestern University). The sequence for SOD1 1–153 was expressed from the vector pET-3d, which was transformed into *E. coli* BL21 (DE3) cells (Novagen, Merck, Darmstadt, Germany). SOD1 protein was purified to homogeneity by Q-Sepharose chromatography as described by Chattopadhyay et al.[23] and Xu et al.[9]. After purification, recombinant wild-type SOD1 was demetalated by dialysis in 10 mM EDTA and 10 mM NaAc buffer (pH 3.8) five times as described by Chattopadhyay et al.[23] and Xu et al.[9]. In all, 10 mM NaAc buffer (pH 3.8) and 20 mM tris-HCl buffer (pH 7.4) were used for further dialysis. The apo SOD1 was then concentrated, filtered, and stored at −80 °C. AAnalyst-800 atomic absorption spectrometer (PerkinElmer) was used to quantify metal content of SOD1 samples. Samples of wild-type SOD1 contained <5% of residual metal ions, indicating that the samples were indeed in the apo state. SDS-PAGE and mass spectrometry were used to confirm that the purified apo SOD1 proteins were single species with an intact disulfide bond. A NanoDrop OneC Microvolume UV-Vis Spectrophotometer (Thermo Fisher Scientific) was used to determine the concentration of apo SOD1 according to its absorbance at 214 nm with a standard calibration curve drawn by BSA.

**SOD1 fibril formation**. Recombinant, full-length apo human SOD1 (30 μM) were incubated in 20 mM tris-HCl buffer (pH 7.4) containing 5 mM TCEP and shaking at 37 °C for 40–48 h, and the SOD1 fibrils were collected. Large aggregates in SOD1 fibril samples were removed by centrifugation for 5000 × *g* at 4 °C for 10 min. The supernatants were then concentrated to ~30 μM in a centrifugal filter (Millipore). A NanoDrop OneC Microvolume UV-Vis Spectrophotometer (Thermo Fisher Scientific) was used to determine the concentrations of the SOD1 fibril according to its absorbance at 214 nm with a standard calibration curve drawn by BSA.

**Congo red binding assays**. SOD1 fibrils were analyzed by Congo red binding assays. A stock solution of 200 μM Congo red was prepared in phosphate-buffered saline and filtered through a filter of 0.22-μm pore size before use. In a typical assay, the SOD1 fibril sample was mixed with a solution of Congo red to yield a

final Congo red concentration of 50 μM and a final SOD1 concentration of 10 μM, and the absorbance spectrum between 400 and 700 nm was then recorded on a Cytation 3 Cell Imaging Multi-Mode Reader (BioTek).

**TEM of SOD1 fibrils**. SOD1 fibrils were examined by TEM of negatively stained samples. Ten microliters of SOD1 fibril samples (~30 μM) were loaded on copper grids for 30 s and washed with $H_2O$ for 10 s. Samples on grids were then stained with 2% (w/v) uranyl acetate for 30 s and dried in air at 25 °C. The stained samples were examined using a JEM-1400 Plus transmission electron microscope (JEOL) operating at 100 kV.

**AFM of SOD1 fibrils**. SOD1 fibrils were produced as described above. Ten microliters of SOD1 fibril samples (~30 μM) were incubated on a freshly cleaved mica surface for 2 min, followed by rinsing three times with 10 μl of pure water to remove the unbound fibrils and drying at room temperature. The fibrils on the mica surface were probed in the air by the Dimension icon scanning probe microscope (Bruker) with ScanAsyst mode. The measurements were realized by using SCANASYST-AIR probe with a spring constant of 0.4 N/m and a resonance frequency of 70 kHz (Bruker). A fixed resolution ($256 \times 256$ data points) of the AFM images was acquired with a scan rate of 1 Hz and analyzed by using NanoScope Analysis 2.0 software (Bruker).

**Cryo-EM of SOD1 fibril**. SOD1 fibrils were produced as described above. An aliquot of 3.5 μl of ~30 μM SOD1 fibril solution was applied to glow-discharged holey carbon grids (Quantifoil Cu R1.2/1.3, 300 mesh), blotted for 3.5 s, and plunge-frozen in liquid ethane using an FEI Vitrobot Mark IV. The grids were examined using an FEI Talos F200C microscope, operated at 200 kV, and equipped with a field emission gun and an FEI Ceta camera (Thermo Fisher Scientific). The cryo-EM micrographs were acquired on an FEI Titan Krios microscope operated at 300 kV (Thermo Fisher Scientific) and equipped with a Gatan Bio-Quantum K2 Summit camera. A total of 2931 movies were collected by beam-image shift data collection methods[51] in super-resolution mode at a nominal magnification of $\times$130,000 (pixel size, 1.04 Å) and a dose of 9.375 $e^-$ $Å^{-2}$ $s^{-1}$ (see Table 1) using SerialEM 3.8.3. An exposure time of 6.4 s was used, and the resulting videos were dose-fractionated into 40 frames. A defocus range of $-1.2$ to $-2.0$ μm was used.

**Helical reconstruction**. All 40 video frames were aligned, summed, and dose-weighted by MotionCor2-1.3.2 and further binned to a pixel size of 1.04 Å (ref. [52]). Contrast transfer function estimation of aligned, dose-weighted micrographs was performed by CTFFIND4.1.8 (ref. [53]). Subsequent image-processing steps, which include manual picking, particle extraction, 2D and 3D classifications, 3D refinement, and post-processing, were performed by RELION-3.1 (ref. [36]).

In total, 10,488 fibrils were picked manually from 2931 micrographs, and 686- and 320-pixel boxes were used to extract particles by 90% overlap scheme. Two-dimensional classification of 686-box size particles was used to calculate the initial twist angle. In regard to helical rise, 4.8 Å was used as the initial value. Particles were extracted into 320-box sizes for further processing. After several iterations of 2D and 3D classifications, particles with the same morphology were picked out. Local searches of symmetry in 3D classification were used to determine the final twist angle and rise value. The 3D initial model was built by selected 2D classes; 3D classification was performed several times to generate a proper reference map for 3D refinement. Three-dimensional refinement of the selected 3D classes with appropriate reference was performed to obtain the final reconstruction. The final map of SOD1 fibril was convergent with a rise of 4.82 Å and a twist angle of $-1.187°$. Postprocessing was performed to sharpen the map with a B factor of $-77.93$ $Å^2$. On the basis of the gold standard Fourier shell correlation (FSC) = 0.143 criteria, the overall resolution was reported as 2.95 Å. The statistics of cryo-EM data collection and refinement are shown in Table 1.

**Atomic model building and refinement**. Coot 0.8.9.2 (ref. [54]) was used to build de novo and modify the atomic model of the SOD1 fibril. The model with three adjacent layers was generated for structure refinement. The model was refined using the real-space refinement program in PHENIX 1.15.2 (ref. [55]). All density map-related figures were prepared in Chimera1.15. Ribbon representation of the structure of SOD1 fibril was prepared in PyMol 2.3.

**Cell viability assays**. SH-SY5Y neuroblastoma cells (catalog number GDC0210) and HEK-293T cells (catalog number GDC0187) were obtained from China Center for Type Culture Collection (CCTCC, Wuhan, China) and cultured in minimum essential media and in Dulbecco's modified Eagle's medium (Gibco, Invitrogen), supplemented with 10% (v/v) fetal bovine serum (Gibco), 100 U/ml streptomycin, and 100 U/ml penicillin in 5% $CO_2$ at 37 °C. SH-SY5Y or HEK-293T cells were plated in 96-well plates in the minimum essential medium. After incubation for 24 h, SOD1 fibril seeds at a final concentration of 0.01, 0.1, 1, or 10 μM were added into the medium for 48 h. The MTT stock solution (5 mg/ml) was diluted with PBS and added to the well for 4 h until formazan was formed in the cells. The final concentration of MTT was 0.5 mg/ml. Finally, the dark blue formazan crystal was dissolved with dimethyl sulfoxide, followed by measuring its absorbance at 492 nm

using a Thermo Multiskan MK3 microplate reader (Thermo Fisher Scientific). Cells were incubated in a medium containing 10% Cell Counting Kit-8 (CCK8) for 2–4 h, and the absorbance of the orange formazan was also measured with a microplate reader at 450 nm. Cell viability was expressed as the percentage ratio of the absorbance of wells containing the treated samples to that of wells containing cells treated with 20 mM Tris-HCl buffer (pH 7.4) containing 5 mM TCEP. The data on cell viability, analyzed by using Origin Pro software version 8.0724 (Origin Laboratory), are expressed as mean ± SD of the values obtained in four or six independent experiments with four different concentrations, and $p$ values were determined using two-sided Student's $t$ test. All experiments were further confirmed by biological repeats.

**Ultrathin section TEM**. SH-SY5Y cells and HEK-293T cells were cultured in 6-well plates in the minimum essential medium for 1 day and then cultured with 0 or 10 μM SOD1 fibril seeds for 3 days, and cells cultured with 20 mM Tris-HCl buffer (pH 7.4) containing 5 mM TCEP as a negative control. After prefixation with 3% paraformaldehyde and 1.5% glutaraldehyde in 1× PBS (pH 7.4), the cells were harvested and postfixed in 1% osmium tetroxide for 1 h using an ice bath; the samples were then dehydrated in graded acetone and embedded in 812 resins. Ultrathin sections of the cells were prepared using a Leica Ultracut S Microtome and negatively stained using 2% uranyl acetate and lead citrate. The doubly stained ultrathin sections of cells were examined using a JEM-1400 Plus transmission electron microscope (JEOL) operating at 100 kV. The TEM images were analyzed by using Origin Pro software version 8.0724 (Origin Laboratory), and $p$ values were determined using two-sided Student's $t$ test. All experiments were further confirmed by biological repeats.

**Reporting summary**. Further information on research design and experimental design is available in the Nature Research Reporting Summary linked to this article.

## Data availability

The cryo-EM density maps have been deposited in the Electron Microscopy Data Bank (EMDB) under accession code EMD-32227 (human SOD1 fibril). The coordinates generated in this study are deposited in the Protein Data Bank (PDB) under accession code PDB 7VZF (human SOD1 fibril). Previously published structure 1HL4 is available from PDB. Biological materials are available on request. Source data are provided with this paper.

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

## Acknowledgements

Y.L. and C.L. acknowledge funding from the National Natural Science Foundation of China (nos. 31770833 and 82188101). Y.L. also acknowledges financial support from the National Natural Science Foundation of China (nos. 32071212 and 31570779), the Key Project of Basic Research, Science and Technology R&D Fund of Shenzhen (no. JCYJ20200109144418639), and the Translational Medicine and Interdisciplinary Research Joint Fund of Zhongnan Hospital of Wuhan University (no. ZNJC201934). C.L. was also supported by the Science and Technology Commission of Shanghai Municipality (nos. 18JC1420500, 20XD1425000, and 2019SHZDZX02), the CAS project for Young Scientists in Basic research (no. YSBR-009), and the Shanghai Pilot Program for Basic Research – Chinese Academy of Sciences, Shanghai Branch (Grant No. CYJ-SHFY-2022-005). L.-Q.W. acknowledges financial support from China Postdoctoral Science Foundation (nos. 2021TQ0252 and 2021M700103) and the Fundamental Research Funds for the Central Universities (no. 2042022kf1047). P.Y. acknowledges financial support from the Major State Basic Research Development Program (no. 2018YFA0507700) and the National Natural Science Foundation of China (no. 31722017). L.Z. acknowledges financial support from the Key Project of Basic Research, Science and Technology R&D Fund of Shenzhen (no. JCYJ20200109144418639). Cryo-EM data were collected at the Center for Biological Imaging, Institute of Biophysics, Chinese Academy of Sciences, China. We thank T. V. O'Halloran (Northwestern University) for the gift of the human SOD1 plasmid; B. Zhu (Center for Biological Imaging, Institute of Biophysics, Chinese Academy of Sciences), X. Li (Center for Biological Imaging, Institute of Biophysics, Chinese Academy of Sciences), L. Wu (Center of Cryo Electron Microscopy, Zhejiang University), and F. Sun (Institute of Biophysics, Chinese Academy of Sciences) for technical assistance with cryo-EM; W. Zou (College of Life Sciences, Wuhan University) for technical assistance with the TEM of ultrathin sections of cells; and Y. Wang (Institute of Biophysics, Chinese Academy of Sciences) for helpful suggestions.

## Author contributions

L.Z., P.Y., C.L., and Y.L. supervised the project. L.-Q.W., C.L., and Y.L. designed the experiments. L.-Q.W., H.-Y.Y., M.-Y.Z., X.H., W.-C.X., and J.C. purified human SOD1

and the SOD1 fibrils. L.-Q.W., H.-Y.Y., and M.-Y.Z. performed Congo red binding assays and cell viability assays of SOD1 fibrils and TEM of ultrathin sections of cells. L.-Q.W., H.-Y.Y., B.D., and Z.W. performed AFM experiments. L.-Q.W., Y.M., H.-Y.Y., K.Z., Q.W., D.Z., and D.L. collected, processed, and/or analyzed cryo-EM data. L.-Q.W., Y.M., C.L., and Y.L. wrote the manuscript. All authors proofread and approved the manuscript.

## Competing interests

The authors declare no competing interests.
