## [Peer Review File · Nature Communications]

REVIEWER COMMENTS

Reviewer #1 (Remarks to the Author):

Wang et al present and describe the first structure of a human SOD1 fibril. SOD1 is involved in the pathogenesis of the motor neuron disease amyotrophic lateral sclerosis where it is observed to form intracellular neuronal aggregates. These proteinaceous inclusions are also observed in cases of Parkinsons disease and aged individuals. Unlike proteins such as a-synuclein or A-beta, SOD1 has been suspected to not form amyloid-like structures due to the behaviour of thioflavin T in aggregation assays and structures of fibrils formed from SOD1 beta-barrel peptides. As such, this is an exceptionally important and long-awaited piece of science that will have far-reaching implications for the field. The paper is concise; conclusions are justified by the results; the methodology is sound (a very similar procedure has recently been used by the group to discover the structure of human prion protein, Wang et al 2020 NSMB Cryo-EM structure of an amyloid fibril formed by full-length human prion protein); and enough information is given to enable repetition. I recommend publication on correction of the few minor points listed below:

Lines 42-51, Please only cite references that directly support the statement made.

Line 55, 'Therefore, it is unclear for the conformational conversion of SOD1 from its mature form into an aggregated form during pathogenesis of ALS.' This sentence should be revised for clarity. As the authors write subsequently, it is very unlikely that the mature, metallated, dimeric and disulphide intact form ever converts into the aggregated form discussed in the paper. The monomer species which polymerises into aggregates is the immature form with no post-translational modifications.

Line 200, As above the authors state 'the gradual reduction of apo SOD1 occurring intracellularly upon loss of metal ions'. Misfolded and aggregated SOD1 evolves from a pool of immature SOD1 rather than conversion of the fully mature state.

Lines 159-162, The authors describe the L-shaped interface as 'mixed compositions of hydrophilic and hydrophobic side chains' and then 'mostly hydrophilic intramolecular interface'. This section needs rewording for clarity and the comment 'whereas side chains of most residues in the interior of such an interface are hydrophilic' should be referenced and/or elaborated upon.

Line 215, The Q153 mutation noted is a synonymous mutation that has no effect on amino acid composition or structure.

Line 431, 'dimer' should be replaced with 'monomer'.

Line 639, The figure legend accompanying Extended Data Fig 2 has been copied from a previous paper (Wang et al 2020 NSMB Cryo-EM structure of an amyloid fibril formed by full-length human prion protein) and should be carefully re-written.

Line 655, add Cys6.

Reviewer #2 (Remarks to the Author):

The manuscript is potentially interesting but shows several limitations:

Major:

1. The term "pathological" as in the title is not justified as it was not established whether the describe fibril, which was purely formed in vitro, shows the same structure as the fibrils in the brain of a patient. It is even less known whether the fibrils are really pathogenic agents in ALS as this disease is thought to depend on SOD oligomers rather than fibrils. The low 'cytotoxicity' of the fibrils in panel Ex. Fig. 1c supports this. Several publications recently showed that the structure of in vitro formed fibrils is not the same as the fibrils in patient tissue. In a revised publication I would expect to see substantial changes in all parts of the manuscript, in particular in the discussion, making it clear that the pathological relevance of the fibril reported in this manuscript is unknown. The MTT assay is also a poor and controversial assay of cytotoxicity. Please change claims of cytotoxicity to mitochondrial dysfunction or provide additional assays of cytotoxic activity, preferably including other cell types.

2. The left-handed structure of the fibril is not sufficiently supported by experimental evidence. The handedness of the structure may thus be wrong. Please determine the handedness with an appropriate method such as AFM or platinum side shadowing.

3. The indentation of the FSC at 0.2 A-1 is indicative of a processing error. Please rework the reconstruction to remove this indentation.

4. The difference spectrum on page 35 cannot have arisen from a subtraction operations described in the figure legend. Addition of the spectra 1, 2 and 3 does not yield spectrum 2.

Minor:

1. The term protofibril is confused in this manuscript with protofilament. The latter is a filamentous substructure of a fibril. The former is a kinetic precursor of mature fibrils.

2. The term structural break, which the authors took from references 37 and 38, is not used appropriately in this manuscript. Structural break was defined previously as structural displacements in the fibril along the z-axis. What the authors mean instead are internal disordered segments; i.e. that the fibril protein forms two or more ordered segments that are interspersed with segments of structural disorder.

3. The term "semi-reducing" is unclear. Either it is reducing or it is not. Semi does not really make sense.

4. Line 99 wrong figure reference. It is part of the extended data.

5. Page 35: What is the "control"?

Reviewer #3 (Remarks to the Author):

Review

The paper by Wang and colleagues builds on publications that addressed SOD1 fibril formation. The Valentine lab at UCLA has extensively studied the fibrils and used EM to

show the fibrils formed from SOD1 and its mutants. The Shaw lab at Baylor also studied the fibrils and showed consistent formation of the fibrils under reducing conditions. The Hart lab at UTHSCSA published a crystal structure of S134N and proposed a mechanism of an intact SOD1 structure forming the fibril (NSB 2003). The Eisenberg lab published the structure of crystals of small stretches of amino acids suspected of forming the aggregates (PNAS 2013). This paper shows, for the first time, the structure of the fibrils at high resolution using Cryo-EM. The structure is significantly different from the WT SOD1 structure and shows clear evidence of a new conformation of SOD1 forming the fibrils. Overall, the work provides an important detailed structure, which was not observed before.

The paper is well written and focused on a topic of interest.

Recommendation:

I recommend that the authors collect structural data on various SOD1 mutants (metal binding region and wild type like mutants). If the mutants showed similar structures this could be a breakthrough to help design molecules to stop the pathogenic aggregation of SOD1

The manuscript is suitable for publication in Nature Communications

Reviewer #1

Remarks to the Author:

Summary:

Wang et al present and describe the first structure of a human SOD1 fibril. SOD1 is involved in the pathogenesis of the motor neuron disease amyotrophic lateral sclerosis where it is observed to form intracellular neuronal aggregates. These proteinaceous inclusions are also observed in cases of Parkinson's disease and aged individuals. Unlike proteins such as alpha-synuclein or Abeta, SOD1 has been suspected to not form amyloid-like structures due to the behaviour of thioflavin T in aggregation assays and structures of fibrils formed from SOD1 beta-barrel peptides. As such, this is an exceptionally important and long-awaited piece of science that will have far-reaching implications for the field. The paper is concise; conclusions are justified by the results; the methodology is sound (a very similar procedure has recently been used by the group to discover the structure of human prion protein, Wang et al 2020 NSMB Cryo-EM structure of an amyloid fibril formed by full-length human prion protein); and enough information is given to enable repetition. I recommend publication on correction of the few minor points listed below.

We sincerely thank the reviewer for recognizing the significance of our work. The reviewer's suggestion is very valuable for us to improve our manuscript. Right now, we have revised and added the following sentences into the revised manuscript, as followed the advice of reviewer #1. *SOD1 is involved in the pathogenesis of the motor neuron disease ALS where it is observed to form intracellular fibrillar inclusions^{3,7,39,40}. These proteinaceous inclusions are also observed in cases of Parkinson's disease and aged individuals^{41,42}. Here, we presented the first structure of a human SOD1 fibril and compared the structures of apo SOD1 dimer and SOD1 fibril produced under reducing conditions (Fig. 4)*

(pages 9-10). Accordingly, four related publications (Refs. 39-42) have been added into the revision.

Minor points

Comment #1 • Lines 42-51, Please only cite references that directly support the statement made.

REPLY: Thanks for the comments. The reviewer is correct. We have now only cited references that directly support the statement made and have adjusted the order of references accordingly, as followed the advice of review #1. We have revised the following sentences in the Introduction section of our article as “*Amyotrophic lateral sclerosis (ALS), also called Lou Gehrig’s disease, is a neurodegenerative disease that involves the loss of upper and lower motor neurons¹⁻⁴. Ninety percent of ALS cases are sporadic and little is known about the origin, while ten percent of ALS cases are inherited familial ALS¹⁻⁷. The sod1 gene, serving as a major antioxidant gene, was the first to be linked to the familial form of ALS⁸ and other genes associated with genetic ALS include those encoding TDP-43 and FUS^{1,6}. The misfolding of human Cu, Zn-superoxide dismutase (SOD1) in motor neuron cells play a crucial role in etiology of the disease^{1,3,4,6,7,9}. Misfolded SOD1 aggregates were widely observed in the spinal cords of both genetic ALS and sporadic ALS cases^{3,4,7}. The functional human SOD1 is a 32-kDa homo-dimeric metalloenzyme; each subunit consists of 153 amino acids and contains one copper ion and one zinc ion^{10,11}. The SOD1 structure in each subunit features an antiparallel fl-barrel composed of eight fl-strands and two α -helices, which is stabilized by a disulfide bond between Cys57 and Cys146 (refs. ^{10,11}). In sharp contrast, the high-resolution structures of SOD1 amyloid fibrils are not available so far^{1,12-14}” (Lines 46-60) and “The mature form of SOD1 is exceptionally stable and it is very unlikely that the mature, metalated, dimeric, and disulfide-intact form ever converts into the aggregated form¹⁵⁻¹⁷. Instead, it has been proposed that immature forms of SOD1, which lack copper and zinc ions and the disulfide bond in the structure, are the origin for cytotoxic misfolded conformations¹⁵⁻¹⁹” (Lines 63-67), as followed reviewer’s suggestion. We have also made sure that reference call-outs are correct.*

Comment #2 • Line 55, ‘Therefore, it is unclear for the conformational conversion of SOD1 from its mature form into an aggregated form during pathogenesis of ALS.’ This sentence should be revised for clarity. As the authors write subsequently, it is very unlikely that the mature, metallated, dimeric and disulphide intact form ever converts into the aggregated form discussed in the paper. The monomer species which polymerises into aggregates is the immature form with no post-translational modifications.

REPLY: Thank for pointing this out. We also think that the sentence ‘Therefore, it is unclear for the conformational conversion of SOD1 from its mature form into an aggregated form during pathogenesis of ALS’ should be revised for clarity. To clarify, we have revised and added the following sentences into the revised manuscript, as followed reviewer’s suggestion. *Therefore, it is unclear for the conformational conversion of SOD1 from its immature form with no post-translational modifications into an aggregated form during pathogenesis of ALS (Lines 60-62). The mature form of SOD1 is exceptionally stable and it is very unlikely that the mature, metalated, dimeric, and disulfide-intact form ever converts into the aggregated form^{15□17} (Lines 63-65).*

Comment #3 • Line 200, As above the authors state ‘the gradual reduction of apo SOD1 occurring intracellularly upon loss of metal ions.’ Misfolded and aggregated SOD1 evolves from a pool of immature SOD1 rather than conversion of the fully mature state.

REPLY: We sincerely thank the reviewer for this suggestion that we should delete the statement “the gradual reduction of apo SOD1 occurring intracellularly upon loss of metal ions”!! We totally agree that misfolded and aggregated SOD1 evolves from a pool of immature SOD1 rather than conversion of the fully mature state. According to the advice of the reviewer, we have now deleted such a statement from the manuscript (Line 224; Line 88). We have revised and added the following sentences into the revised manuscript. *In all three models, SOD1 fibrils are produced from the immature form of the protein under reducing conditions (Lines 223-224). In our SOD1 fibril model, Cys57, Cys111, and Cys146 are all in disulfide-reduced conformations with free thiol groups, and misfolded and aggregated SOD1 evolves from a pool of immature*

SOD1 (Lines 226-228). *Treatment of the apo form of SOD1 with 5□10 mM tris (2-carboxyethyl) phosphine (TCEP), a highly stable disulfide-reducing agent, generates a reduced state that is able to mimic physiological reducing environments* (Lines 85-88).

Comment #4 • Lines 159-162, The authors describe the L-shaped interface as ‘mixed compositions of hydrophilic and hydrophobic side chains’ and then ‘mostly hydrophilic intramolecular interface’. This section needs rewording for clarity and the comment ‘whereas side chains of most residues in the interior of such an interface are hydrophilic’ should be referenced and/or elaborated upon.

REPLY: We apologize for this confusion. We totally agree that this section needs rewording for clarity and the comment ‘whereas side chains of most residues in the interior of such an interface are hydrophilic’ should be referenced and/or elaborated upon. To clarify and elaborate upon, we have reworded and added the following sentence into the revised manuscript. *Side chains of most residues (Lys36, Thr39, His43, His46, His48, Ser98, Glu100, Asp101, Ser105, Ser107, and Asp109) in the interior of the intramolecular L-shaped interface are hydrophilic (Figs. 2b,g and 3a)* (Lines 168-170).

Comment #5 • Line 215, The Q153 mutation noted is a synonymous mutation that has no effect on amino acid composition or structure.

REPLY: Thanks the reviewer for the comments. Yes, the Q153 mutation noted is a synonymous mutation that has no effect on amino acid composition or structure. According to the advice of the reviewer, we have now deleted the term “Gln153” from the manuscript (Line 239) and have revised the following sentence in the Discussion section of the revised manuscript. *Notably, residues forming strong salt bridges (...) that contribute to stabilization of the intramolecular L-shaped interface between the N- and C-terminal parts of SOD1 fibril (Figs. 2a,b,f,g and 3b□e) or hydrogen bonds (Val14 and Asp125) that contribute to maintenance of the SOD1 fibril structure (Extended Data Fig. 7b□d) are also ALS-associated mutation sites^{1□5,8,17□22,25,26,29,44□50}* (Lines 235-240).

Comment #6 • Line 431, ‘dimer’ should be replaced with ‘monomer’.

REPLY: Thank for pointing this out. Right now, we have replaced ‘dimer’ with ‘monomer’ in our manuscript (Line 471, Legend of Fig. 4), as followed reviewer’s suggestion.

Comment #7 • Line 639, The figure legend accompanying Extended Data Fig. 2 has been copied from a previous paper (Wang et al 2020 NSMB Cryo-EM structure of an amyloid fibril formed by full-length human prion protein) and should be carefully re-written.

REPLY: We apologize for this mistake. We have now carefully rewritten the figure legend accompanying Extended Data Fig. 3, as followed reviewer’s suggestion. *Extended Data Fig. 3 Cryo-EM images of SOD1 fibril. a Reference-free 2D class averages of SOD1 fibril showing two protofilaments intertwined. Scale bar, 10 nm. b Enlarged image of (a) showing two protofilaments arranged in a staggered manner. Scale bar, 2 nm (page 42, Legend of Extended Data Fig. 3).*

Comment #8 • Line 655, add Cys6.

REPLY: Thanks for the comments. The reviewer is correct. As suggested by the reviewer, we have now added Cys6 into the figure legend accompanying Extended Data Fig. 6. *Hydrophobic residues and sulfur atoms in Cys6, Cys111, and Cys146 are highlighted in orange and yellow, respectively, and five hydrophobic cavities (i to v) in the SOD1 fibril are magnified in b to f (page 45, Legend of Extended Data Fig. 6).*

Reviewer #2

Remarks to the Author:

Summary:

The manuscript is potentially interesting but shows several limitations.

We sincerely thank the reviewer for recognizing the significance of our work.

The reviewer’s suggestion is very valuable for us to improve our manuscript.

Major points

Comment #1 • The term “pathological” as in the title is not justified as it was not established whether the describe fibril, which was purely formed in vitro, shows the same structure as the fibrils in the brain of a patient. It is even less known whether the

fibrils are really pathogenic agents in ALS as this disease is thought to depend on SOD oligomers rather than fibrils. The low 'cytotoxicity' of the fibrils in panel Ex. Fig. 1c supports this. Several publications recently showed that the structure of in vitro formed fibrils is not the same as the fibrils in patient tissue. In a revised publication I would expect to see substantial changes in all parts of the manuscript, in particular in the discussion, making it clear that the pathological relevance of the fibril reported in this manuscript is unknown. The MTT assay is also a poor and controversial assay of cytotoxicity. Please change claims of cytotoxicity to mitochondrial dysfunction or provide additional assays of cytotoxic activity, preferably including other cell types.

REPLY: Thank the reviewer for this great point! Indeed, the term “pathological” as in the title is not justified as it was not established whether the describe fibril, which was purely formed in vitro, shows the same structure as the fibrils in the brain of a patient. Moreover, it is still unclear whether the fibrils are pathogenic agents in ALS. According to the reviewer’s suggestion, we have now deleted the term “pathological” from the title and have used the term “*physiological and fibrillar states*” instead of “physiological and pathological states” throughout the revised manuscript. Thank the reviewer #2’s comments regarding the pathological significance of the in vitro fibrils here reported. We totally agree that the work should be revised by addressing in the discussion the pathological relevance in detail and clearly stating that the pathological relevance remains unknown. We have clearly stated that the pathological relevance remains unknown in the Discussion section of the revision, in order to address the major concern from reviewer #2. *Cryo-EM structure of an amyloid fibril formed by full-length human SOD1 reveals its conformational conversion* (page 1, the Title). *Our findings provide structural insights into the conversion of SOD1 between physiological and fibrillar states* (Lines 81-82). *Notably, the SOD1 molecule adopts largely distinctive..., highlighting the high structural polymorphs and phenotypic diversity of SOD1 in physiological and fibrillar states* (Lines 179-182). *Of note, the pathological relevance of the SOD1 fibril reported here remains unknown, although treatment of cells with the fibrils disrupts mitochondrial membrane permeability and integrity³¹, causes severe mitochondrial impairment (this work) and inflammation³², and activates microglia³³. It is still unclear whether the fibrils are pathogenic agents in ALS^{9,13} (pages 10-11, the Discussion section).*

We sincerely thank reviewer #2 for this important suggestion that we should measure the cytotoxic activity of SOD1 fibrils to SH-SY5Y cells and other cell types using another cytotoxicity assay!! We also sincerely thank the reviewer for his (her) expert suggestion that we should assess the mitochondrial dysfunction caused by SOD1 fibrils!! According to the advice of reviewer #2, we have now provided the cytotoxicity data of SOD1 fibrils to SH-SY5Y and HEK-293T cells assessed by two assays (Extended Data Fig. 1c-f, page 38) and mitochondrial dysfunction data of SOD1 fibril-treated cells examined by ultrathin section TEM (Extended Data Fig. 2, page 40). We have revised and added the following sentences into the Results section and the Methods section of the revision. *Notably, the SOD1 fibrils exhibited cytotoxicity to both SH-SY5Y cells (Extended Data Fig. 1c,d) and HEK-293T cells (Extended Data Fig. 1e,f) in a dose-dependent manner, and caused severe mitochondrial impairment in both cell lines (Extended Data Fig. 2a-j). Together, these data showed that full-length apo SOD1 forms cytotoxic, mitochondrial dysfunction-inducing amyloid fibrils under reducing conditions (Lines 102-107). Cells were incubated in new medium containing 10% Cell Counting Kit-8 (CCK8) for 2 to 4 hours, and the absorbance of the orange formazan was also measured with a microplate reader at 450 nm (pages 33-34). A subsection titled “Ultrathin section TEM” has been added into Methods section of the revision (page 34).*

Extended Data Fig. 1c-f SOD1 forms cytotoxic amyloid fibrils under reducing conditions. **c, d** Cytotoxicity of SOD1 amyloid fibril seeds to SH-SY5Y cells assessed by the MTT assay (**c**) and the CCK8 assay (**d**). **e, f** Cytotoxicity of SOD1 amyloid fibril seeds to HEK-293T cells assessed by the MTT assay (**e**) and the CCK8 assay (**f**). Cells were treated with indicated concentrations of SOD1 fibril seeds for 2 days. Cell viability data were normalized to cells treated with 20 mM Tris-HCl buffer (pH 7.4) containing 5 mM TCEP (gray bar), and are expressed as mean \pm S.D. of the values obtained in four (**c, e**) or six (**d, f**) independent experiments. 0.01, 0.1, 1, and 10 μ M SOD1 fibril seeds, (**c**) $p = 0.085, 0.10, 0.0034,$ and $0.00058,$ respectively, (**d**) $p = 0.67, 0.042, 0.00043,$ and $0.0000013,$ respectively, (**e**) $p = 0.46, 0.057, 0.0017,$ and $0.000083,$ respectively, and (**f**) $p = 0.71, 0.42, 0.00057,$ and $0.000018,$ respectively. SH-SY5Y or HEK-293T cells treated with 20 mM Tris-HCl buffer (pH 7.4) containing 5 mM TCEP for 2 days were used as a control. Statistical analyses were performed using the Student's *t*-test. Values of $p < 0.05$ indicate statistically significant differences. The following notation is used throughout: * $p < 0.05$; ** $p < 0.01$; *** $p < 0.001$; and **** $p < 0.0001$ relative to the control. Data behind graphs are available as source data.

Extended Data Fig. 2 Treatment of cells with SOD1 amyloid fibril seeds causes severe mitochondrial impairment. **a-h** SH-SY5Y cells (**a-d**) and HEK-293T cells (**e-h**) were cultured for 1 day and then incubated with 0 μ M SOD1 fibril seeds (**a, b, e,** and **f**) or 10 μ M SOD1 fibril seeds (**c, d, g,** and **h**) for 3 days. The enlarged regions (**b, d, f,** and **h**) show 12-fold enlarged images from (**a, c, e,** and **g**), respectively, and display the detailed structures of mitochondria in cells. Nuclei are highlighted using black arrows (**a, c, e,** and **g**). The morphology of normal mitochondria in SH-SY5Y (**b**) and HEK-293T (**f**) cells incubated with 0 μ M SOD1 fibril seeds, which are highlighted by blue arrows, was tubular or round. 10 μ M SOD1 fibril seed treatment caused severe mitochondrial impairment in SH-SY5Y (**d**) and HEK-293T (**h**) cells. Most of the mitochondria in the cells (<50%) became swollen and vacuolized, which is highlighted by red arrows. Samples were negatively stained using 2% uranyl acetate

and lead citrate. The scale bars represent 2 μm (**a**, **c**, **e**, and **g**) and 500 nm (**b**, **d**, **f**, and **h**), respectively. **i**, **j** Quantification of TEM images performed on biological replicates show that SOD1 amyloid fibrils significantly increase mitochondrial damage in SH-SY5Y (**i**) and HEK-293T (**j**) cells. In box plots, the center of the box represents the median value, the box represents the 25th to 75th percentile with the median, and the whiskers represent the minimum to the maximum range apart from outliers. About 30 cells were counted in each group. SH-SY5Y or HEK-293T cells treated with 20 mM Tris-HCl buffer (pH 7.4) containing 5 mM TCEP for 3 days were used as a control. A significantly lower number of normal mitochondria was observed in SOD1 fibril-treated cells than did in control cells treated by Tris-HCl buffer containing TCEP (**i**, $p = 0.000000000014$, and **j**, $p = 0.000000000048$). Statistical analyses were performed using the Student's t -test. Values of $p < 0.05$ indicate statistically significant differences. The following notation is used throughout: $*p < 0.05$; $**p < 0.01$; $***p < 0.001$; and $****p < 0.0001$ relative to control.

Comment #2 • The left-handed structure of the fibril is not sufficiently supported by experimental evidence. The handedness of the structure may thus be wrong. Please determine the handedness with an appropriate method such as AFM or platinum side shadowing.

REPLY: We sincerely thank the reviewer for this important suggestion that we should determine the handedness of SOD1 fibrils with an appropriate method such as AFM!! Indeed, the left-handed structure of the fibril is not sufficiently supported by experimental evidence. The handedness of the structure may thus be wrong. According to the advice of reviewer #2, we have now provided high-resolution AFM images of SOD1 fibrils (Extended Data Fig. 4, page 43) and have revised and added the following sentences into the Results section of the revision. *The cryo-EM micrographs, two-dimensional (2D) class average images, and atomic force microscopy (AFM) images show that the SOD1 fibril is composed of a single protofilament with a left-handed helical twist (Fig. 1a, Extended Data Fig. 3a, and Extended Data Fig. 4a–e). The helical pitch is 144 ± 5 nm (Fig. 1a) or 146 ± 5 nm (Extended Data Fig. 4a–e). The SOD1 subunit within the protofilament is arranged in a staggered manner (Extended Data Fig. 3b). The fibrils are morphologically homogeneous, showing a fibril full width of 12.3 ± 0.7 nm (Fig. 1a and Extended*

Data Fig. 3a) or 12.9 ± 1.0 nm (Extended Data Fig. 4a-e) (pages 6-7). The 3D map showed a single protofilament in the SOD1 fibril with a left-handed helix, and the left-handed structure of the fibril is supported by AFM images (Extended Data Fig. 4a-e) (Lines 124-127). A subsection titled “AFM of SOD1 fibrils” has been added into Methods section of the revision (page 31).

Extended Data Fig. 4 High-resolution AFM images of SOD1 fibrils. **a**, **c**, and **e** AFM images of amyloid fibrils assembled from recombinant, full-length apo human SOD1 under reducing conditions. The enlarged regions (**b** and **d**) show 16-fold enlarged images from (**a** and **c**, red square), respectively, and visualize the detailed morphology of SOD1 fibrils. Enlarged sections of **b**, **d**, and **e** (right) showing the SOD1 fibril intertwined into a left-handed helix, with a fibril full width of 12.9 ± 1.0 nm ($n = 8$) and a helical pitch of 146 ± 5 nm ($n = 8$). The scale bars represent 500 nm (**a**, **c**, and **e**) and 100 nm (**b** and **d**), respectively. The helical pitch and fibril width were measured and expressed as the mean \pm SD of values obtained in eight independent measurements.

Comment #3 • The indentation of the FSC at 0.2 \AA^{-1} is indicative of a processing error. Please rework the reconstruction to remove this indentation.

REPLY: We sincerely thank the reviewer for this important suggestion!! We totally agree that the indentation of the FSC at 0.2 \AA^{-1} is indicative of a

processing error. According to the reviewer's suggestion, we have reworked the reconstruction to remove this indentation. *The reconstruction was reworked and gold-standard refinement was used for estimation of the density map resolution. The global resolution of 2.95 Å was calculated using a Fourier shell correlation (FSC) curve cut-off at 0.143* (Extended Data Fig. 5 and the Legend, page 44). We re-generated a new density map with a higher resolution (2.95 Å instead of 3.08 Å) and re-uploaded this new density map and the corresponding FSC curve to Protein Data Bank on March 1, 2022. Our updated PDB entry ID 7VZF and EMDB entry ID EMD-32227 were approved by Protein Data Bank on March 24, 2022. We have also provided an updated Table 1 (page 28) and an updated Validation report. We have revised and added the following sentences into the Results section of the revision. *Using helical reconstruction in RELION3.1 (ref. ³⁶), we determined a density map of the ordered core of SOD1 fibril, with an overall resolution of 2.95 Å, which features well-resolved side-chain densities and clearly separated β strands along the fibril axis (Fig. 1b and Extended Data Fig. 5) (Lines 119-122). We unambiguously built a structure model of SOD1 fibril comprising the N-terminal segment (residues 3 to 55) and the C-terminal segment (residues 86 to 153) at 2.95 Å (Fig. 2) (Lines 130-132). Postprocessing was performed to sharpen the map with a B factor of -77.93 \AA^2 . On the basis of the gold standard Fourier shell correlation (FSC) = 0.143 criteria, the overall resolution was reported as 2.95 Å (pages 32-33).*

Extended Data Fig. 5 Global resolution estimate for the SOD1 fibril reconstructions. The reconstruction was reworked and gold-standard refinement was

used for estimation of the density map resolution. The global resolution of 2.95 Å was calculated using a Fourier shell correlation (FSC) curve cut-off at 0.143.

Table 1 Cryo-EM data collection, refinement and validation statistics

	SOD1 fibril (EMD-32227, PDB 7VZF)
Data collection and processing	
Magnification	130,000
Voltage (kV)	300
Camera	K2 summit (Titan Krios)
Frame exposure time (s)	0.16
Movie frames (n)	40
Electron exposure (e ⁻ /Å ²)	60
Defocus range (µm)	□2.0 to □1.2
Pixel size (Å)	1.04
Symmetry imposed	C1
Box size (pixel)	320
Inter-box distance (Å)	33.3
Micrographs collected (n)	2,931
Segments extracted (n)	288,744
Segments after Class2D (n)	147,525
Segments after Class3D (n)	70,067
Map resolution (Å)	2.95
FSC threshold	0.143
Map resolution range (Å)	2.30□5.01
Refinement	
Initial model used	De novo
Model resolution (Å)	2.95
FSC threshold	0.143
Model resolution range (Å)	2.95
Map sharpening B factor (Å ²)	□77.93
Model composition	
Nonhydrogen atoms	2,628
Protein residues	363
Ligands	0
B factors (Å ²)	
Protein	70.90
R.m.s. deviations	
Bond lengths (Å)	0.009
Bond angles (°)	1.060
Validation	
MolProbity score	2.86
Clashscore	37.29
Poor rotamers (%)	0
Ramachandran plot	
Favored (%)	73.50
Allowed (%)	26.50
Disallowed (%)	0

Comment #4 • The difference spectrum on page 35 cannot have arisen from a subtraction operation described in the figure legend. Addition of the spectra 1, 2 and 3 does not yield spectrum 2.

REPLY: We have carefully checked the difference spectra data in Extended Data Fig. 1b (page 38), as followed reviewer's suggestion. It is correct. Source Data for Extended Data Fig. 1b are provided with this paper. We have revised and added the following sentence into the revised manuscript. *The absorbance spectrum 2 (SOD1 fibril + Congo red, green) minus the absorbance spectrum 1 (Congo red alone, red) with the maximum absorbance at 490 nm minus the absorbance spectrum 3 (SOD1 fibril alone, black) equals the difference spectrum 4 (blue) with the maximum absorbance at 550 nm* (Lines 685-689, Legend of Extended Data Fig. 1).

Minor points

Comment #1 • The term protofibril is confused in this manuscript with protofilament. The latter is a filamentous substructure of a fibril. The former is a kinetic precursor of mature fibrils.

REPLY: We apologize for this confusion. Indeed, the term protofibril is confused in the previous version of our manuscript with protofilament. The latter is a filamentous substructure of a fibril, but the former is a kinetic precursor of mature fibrils. We have now used the term “*protofilament*” instead of “*protofibril*” throughout the revised manuscript, as followed reviewer's suggestion. The followings are examples. *The SOD1 fibril consists of a single protofilament with a left-handed helix* (page 3, the fifth sentence of the Abstract). *The SOD1 fibril is composed of a single protofilament with a fibril full width of 11.3 ± 1.0 nm ($n = 8$)* (Extended Data Fig. 1a)... (Lines 96-97). *The cryo-EM micrographs, two-dimensional (2D) class average images, and atomic force microscopy (AFM) images show that the SOD1 fibril is composed of a single protofilament with a left-handed helical twist* (Fig. 1a, Extended Data Fig. 3a, and Extended Data Fig. 4a-e). *The helical pitch is 144 ± 5 nm* (Fig. 1a) *or 146 ± 5 nm* (Extended Data Fig. 4a-e). *The SOD1 subunit within the protofilament is arranged in a staggered manner* (Extended Data Fig. 3b) (pages 6-7). *Cross-sectional view of the 3D map of the SOD1 fibril and top view of the density map show a protofilament*

comprising the N- and the C-terminal segments, with an unstructured flexible region in between (Fig. 1b,e). The 3D map showed a single protofilament in the SOD1 fibril with a left-handed helix, and the left-handed structure of the fibril is supported by AFM images (Extended Data Fig. 4a–e). The half-helical pitch is 73.1 nm (Fig. 1c). The SOD1 subunit within the protofilament stacks along the fibril axis with a helical rise of 4.82 Å and twist of -1.187° (Fig. 1d) (Lines 122-129).

Comment #2 • The term structural break, which the authors took from references 37 and 38, is not used appropriately in this manuscript. Structural break was defined previously as structural displacements in the fibril along the z-axis. What the authors mean instead are internal disordered segments; i.e. that the fibril protein forms two or more ordered segments that are interspersed with segments of structural disorder.

REPLY: We sincerely thank the reviewer for the comments. The reviewer is correct. Indeed, the term structural break, which we took from references 37 and 38, is not used appropriately in the previous version of our manuscript. According to the reviewer’s suggestion, we have now used the terminology “*internal disordered segment*” or “*intrinsic disordered segment*” instead of “structural break” throughout the revised manuscript, as followed reviewer’s suggestion. *The fibril core exhibits a serpentine fold comprising N-terminal segment (residues 3 to 55) and C-terminal segment (residues 86 to 153) with an intrinsic disordered segment (page 3, the sixth sentence of the Abstract). The density of an intrinsic disordered segment comprising residues 56 to 85 is invisible due to high flexibility (Fig. 2a–c), which is reminiscent of the internal disordered segments observed in the structures of patient-derived amyloid fibrils from systemic AL amyloidosis^{37,38}. The presence of the internal disordered segment represents an interesting structural feature of SOD1 fibril formed under reducing conditions (pages 7-8). Moreover, the cytotoxic SOD1 fibril structure features a long, mostly hydrophilic intramolecular L-shaped interface and an intrinsic disordered segment comprising residues 56 to 85 (Fig. 4a,c). Once apo SOD1 dimer converts into its fibrillar form, the SOD1 molecule undergoes a completely conformational rearrangement, with the antiparallel β -barrel of apo SOD1 converted to β 1– β 5, β 7, β 8, β 11, and β 13, the loop between β 4’ and α 1 converted to β 6, α 1 and the loop between α 1 and β 5’ converted to the internal disordered segment, ... (Lines 191-*

198). *The SOD1 fibril displays a very compact fold with an internal disordered segment...(Lines 260-261).*

Comment #3 • The term “semi-reducing” is unclear. Either it is reducing or it is not. Semi does not really make sense.

REPLY: We sincerely thank the reviewer for the comments. The reviewer is correct. We totally agree that the term “semi-reducing” is unclear and semi does not really make sense. According to the advice of reviewer #2, we have now used the term “*reducing*” instead of “semi-reducing” throughout the revised manuscript. The followings are examples. *Here we produced cytotoxic amyloid fibrils from full-length apo human SOD1 under reducing conditions and determined the atomic structure using cryo-EM (page 3, the fourth sentence of the Abstract). SOD1 forms cytotoxic amyloid fibrils under reducing conditions. Treatment of the apo form of SOD1 with 5–10 mM tris (2-carboxyethyl) phosphine (TCEP), a highly stable disulfide-reducing agent, generates a reduced state that is able to mimic physiological reducing environments (Lines 85-88). SOD1 fibrils formed under such reducing conditions were concentrated to ~30 μ M in a centrifugal filter (Millipore) and examined by electron microscopy without further treatment (Lines 91-93). Negative-staining transmission electron microscopy (TEM) imaging showed that recombinant, full-length apo SOD1 formed homogeneous and unbranched fibrils under reducing conditions (Extended Data Fig. 1a). The SOD1 fibril is composed of a single protofilament with a fibril full width of 11.3 ± 1.0 nm ($n = 8$) (Extended Data Fig. 1a), which is consistent with previously described in vitro amyloid fibrils produced from full-length apo SOD1 under the same reducing conditions... (Lines 94-99). Together, these data showed that full-length apo SOD1 forms cytotoxic, mitochondrial dysfunction-inducing amyloid fibrils under reducing conditions (Lines 105-107). The presence of the internal disordered segment represents an interesting structural feature of SOD1 fibril formed under reducing conditions (Lines 135-137). The presence of a mostly hydrophilic intramolecular interface represents another interesting structural feature of SOD1 fibril formed under reducing conditions (Lines 171-172). Here, we presented the first structure of a human SOD1 fibril and compared the structures of apo SOD1 dimer and SOD1 fibril produced under reducing conditions (Fig. 4) (pages 9-10). In all three models,*

SOD1 fibrils are produced from the immature form of the protein under reducing conditions (Lines 223-224).

Comment #4 • Line 99 wrong figure reference. It is part of the extended data.

REPLY: Thanks the reviewer for the comment. The reviewer is correct. Indeed, Line 99 wrong figure reference, and it is part of the extended data. We have now revised and added the following sentence into the revised manuscript, as followed the advice of review #2. *Notably, the SOD1 fibrils exhibited cytotoxicity to both SH-SY5Y cells (Extended Data Fig. 1c,d) and HEK-293T cells (Extended Data Fig. 1e,f) in a dose-dependent manner, and caused severe mitochondrial impairment in both cell lines (Extended Data Fig. 2a-j) (Lines 102-105).*

Comment #5 • Page 35: What is the “control”?

REPLY: To address the question from reviewer #2, we have now revised and added the following sentence into the Legend of Extended Data Fig. 1 of the revision. *SH-SY5Y or HEK-293T cells treated with 20 mM Tris-HCl buffer (pH 7.4) containing 5 mM TCEP for 2 days were used as a control (Lines 700-702).*

Reviewer: 3

Remarks to the Author:

Review:

The paper by Wang and colleagues builds on publications that addressed SOD1 fibril formation. The Valentine lab at UCLA has extensively studied the fibrils and used EM to show the fibrils formed from SOD1 and its mutants. The Shaw lab at Baylor also studied the fibrils and showed consistent formation of the fibrils under reducing conditions. The Hart lab at UTHSCSA published a crystal structure of S134N and proposed a mechanism of an intact SOD1 structure forming the fibril (NSB 2003). The Eisenberg lab published the structure of crystals of small stretches of amino acids suspected of forming the aggregates (PNAS 2013). This paper shows, for the first time, the structure of the fibrils at high resolution using Cryo-EM. The structure is significantly different from the WT SOD1 structure and shows clear evidence of a new conformation of SOD1 forming the fibrils. Overall, the work provides an important detailed structure, which was not observed before.

The paper is well written and focused on a topic of interest.

The manuscript is suitable for publication in Nature Communications.

We sincerely thank the reviewer for recognizing the significance of our work. The reviewer's suggestion is very valuable for us to improve our manuscript. Right now, we have added the following paragraph into the revised manuscript, as followed the advice of review #3. *This work builds on previous intensive studies of SOD1 fibril formation^{12,14,19,23,30,43}. Valentine and colleagues extensively investigated SOD1 fibrils and used EM and AFM to characterize the fibrils formed by SOD1 and its mutants^{23,26}. Shaw and co-workers also studied SOD1 fibrils and showed consistent formation of the fibrils under reducing conditions³⁰. The Hart laboratory determined the crystal structures of two pathogenic SOD1 mutants S134N and H46R, termed metal-binding region mutants, and proposed a mechanism of amyloid structures assembled from S134N (or H46R) dimers⁴³. The Eisenberg group reported the crystal structures of several key fibril-forming segments of SOD1¹⁴ (Lines 204-212). Accordingly, one related publication (Ref. 43) has been added into the revision.*

Recommendation:

Comment #1 • I recommend that the authors collect structural data on various SOD1 mutants (metal binding region and wild type like mutants). If the mutants showed similar structures this could be a breakthrough to help design molecules to stop the pathogenic aggregation of SOD1.

REPLY: Thank the reviewers #3's recommendation regarding the collection of structural data on various SOD1 mutants (metal-binding region and wild-type-like mutants). We totally agree that if the mutants showed similar structures this could be a breakthrough to help design molecules to stop the pathogenic aggregation of SOD1. According to the advice of the reviewer, we have now added the following discussion into the revised manuscript. *Thus, we plan to collect structural data on various SOD1 mutations including metal-binding region mutants H46R, H46D, G85R, and D125H and wild-type-like mutants A4V, D90A, and G93A in the near future (Lines 247-250, Discussion).* Accordingly, one related publication (Ref. 50) has been added into the revision.

REVIEWERS' COMMENTS

Reviewer #1 (Remarks to the Author):

The authors have addressed my comments, made improvements to cell viability assays and included AFM data.

Reviewer #2 (Remarks to the Author):

The authors constructively and satisfactorily addressed all concerns that I raised previously.

Reviewer #1

Remarks to the Author:

The authors have addressed my comments, made improvements to cell viability assays and included AFM data.

We sincerely thank the reviewer for recognizing the significance of our work. The reviewer's suggestion is very valuable for us to improve our manuscript.

Reviewer #2

Remarks to the Author:

The authors constructively and satisfactorily addressed all concerns that I raised previously.

We sincerely thank the reviewer for recognizing the significance of our work. The reviewer's suggestion is very valuable for us to improve our manuscript.